# Interlaboratory Concordance of a Multiplex ELISA for Lyme and Lyme-like Illness Using Australian Samples and Commercial Reference Panels: A Proof-of-Concept Study

**DOI:** 10.3390/pathogens14121281

**Published:** 2025-12-12

**Authors:** Kunal Garg, Fausto Villavicencio-Aguilar, Flora Solano-Rivera, Leona Gilbert

**Affiliations:** 1Tezted Ltd., Mattilaniemi 6-8, 40100 Jyväskylä, Finland; kunal.garg@tezted.com; 2Sanoviv Medical Institute, KM 39 Carretera Libre Tijuana-Ensenada s/n Interior 6, Playas de Rosarito 22712, Baja California, Mexico; fausto.villavicencio@sanoviv.com (F.V.-A.); flora.solano@sanoviv.com (F.S.-R.)

**Keywords:** Australia, Lyme borreliosis, tick, tick-borne disease, Lyme-like disease, Tickplex

## Abstract

Tick bites acquired in the northern or southern hemisphere can transmit microbes that may cause illness. The most prevalent infection is Lyme borreliosis (LB), with all proven cases to date having been acquired in the northern hemisphere. The existence of endemic LB in Australia has not been proven explicitly, and there is uncertainty concerning the cause of “Lyme-like” disease (LLD) in Australia. As many tick-borne diseases (TBDs) are diagnosed by serology, validated assays for use in both the northern and southern hemispheres are required. Using a multiplex enzyme-linked immunosorbent assay (TICKPLEX^®^), two independent laboratories tested a total of 53 well-characterized reference sera that consisted of 33 samples from northern hemisphere patients with confirmed tick-borne disease (TBD) and 20 randomly selected sera from Australian patients with suspected TBDs, presenting with or without LLD. Antibody responses to multiple microbial antigens from causative agents of TBDs were found. High concordance between laboratories was demonstrated on this small set of samples. The results obtained provide the basis for further evaluation of TICKPLEX^®^ on a larger number of samples from Australian patients with suspected TBDs. These findings should be considered preliminary, providing proof-of-concept evidence that warrants validation in larger, clinically diverse cohorts.

## 1. Introduction

Globally, after mosquitoes, ticks are the second most common blood-sucking parasites that transmit infectious agents to humans, with the most widespread tick-borne disease (TBD) being Lyme disease [1,2] or Lyme borreliosis (LB). Infection with *Borrelia burgdorferi* sensu lato (Bb s.l.), the causative agent of LB, is the most frequently reported TBD globally. Dong et al. reported that seroprevalence of *Borrelia burgdorferi* varies substantially across regions, with the highest rates observed in Europe and Asia [2]. These estimates are based primarily on data from endemic regions and may not reflect the global situation. Limited surveillance in non-endemic areas continues to constrain accurate worldwide burden estimates [2]. For 2022, the state and local health departments in the United States reported 62,551 LB cases to the Centers for Disease Control and Prevention (CDC) [3]. These numbers may be a gross underestimate, as suggested by other studies. For example, for LB alone, the actual number is between 3- and 12-fold higher than reported [4,5]. The existence of endemic LB in Australia has not been proven, and there is uncertainty concerning the cause of “Lyme-like” disease (LLD) in Australia. This has been the subject of debate for over 30 years. Spirochaetes of the Bb complex, which includes those that cause LB, have been identified in Australian patients and ticks [6,7,8,9,10], but conclusive evidence has not been provided according to other studies [11,12,13,14]. Nevertheless, Australian patients, some of whom have not travelled overseas [7,10], have been identified with LLD and have symptoms consistent with LB [15,16,17,18,19,20,21].

The incidence of LLD is likely underestimated in Australia due to a lack of public health surveillance programs for TBD, including LLD. Despite many reports of patients suffering severe illness following a tick bite, many such case reports are dismissed [19,22]. In 2018, the Australian Government proposed the diagnostic label ‘Debilitating Symptom Complexes Attributed to Ticks’ (DSCATT) for a chronic syndrome of unclear etiology associated with tick bites [23]. However, to date, the number of cases of TBDs remains unclear. The evidence that describes the relationship between tick bites and DSCATT in Australia is limited to only one publication with a sample size of 29 patients [24]. Anecdotal reports from general practitioners, local governments, and concerned community members indicate that TBD is a significant public health issue [22].

This diagnostic uncertainty was highlighted in the 2025 Australian Senate inquiry, which concluded that the DSCATT pathway lacks clinical utility and contributes to delays in care. The Committee acknowledged the substantial symptom burden reported by patients and emphasized the need for improved diagnostics and updated clinical guidelines in Australia [25]. These policy findings underscore the importance of evaluating broader diagnostic tools such as multiplex serological assays. Also, the lack of progress in surveillance and diagnostic capability may partly explain why only a limited number of TBDs are recognized as endemic in Australia: Q fever (*Coxiella burnetii*) and spotted fevers caused by *Rickettsia australis* and *Rickettsia honei* [26]. Additionally, there is concern about unidentified TBDs in Australia (i.e., not caused by Rickettsia and Coxiella), such that a broader range of testing is required for other microbial causes of TBD [26].

For LB, the diagnosis of LLD may still be a clinical diagnosis. A history of tick exposure (or travel to an endemic area) and secondary or tertiary manifestations may suggest LB. However, the differential diagnosis includes aseptic meningitis, Bell’s palsy, peripheral neuropathy, multiple sclerosis, acute and chronic arthritis [27], and, in Australia, arbovirus infections such as epidemic polyarthritis (Ross River), and well-known TBD such as tick typhus (spotted fever) [28,29,30,31,32,33]. Queensland tick typhus (QTT), the classic Australian spotted fever caused by *Rickettsia australis*, is a common illness in the Northern Beaches Council area following a tick bite, causing fever, rash, and characteristic skin lesions (often vesicular or even pustular), usually with an eschar at the bite site. QTT is occasionally fatal, but early recognition and implementation of appropriate antibiotic treatment usually leads to rapid resolution of illness. A confirmed diagnosis of QTT is based on immunofluorescent antibody (IFA) and specific *R. australis* polymerase chain reaction (PCR) tests. A confirmed diagnosis of LB is made by either PCR or culture of the causative spirochaete from patient samples, mainly skin biopsies. However, culture of Borrelia is impractical for routine diagnosis, and PCR—though useful—has limited availability. Accordingly, serology remains the primary diagnostic approach, traditionally involving IFA or ELISA with confirmatory Western blot. More recently, modified two-tier ELISA algorithms have demonstrated comparable accuracy to ELISA/WB approaches [34]. Against this policy backdrop, the broader epidemiology of endemic and imported TBDs in Australia requires continued scientific investigation.

These uncertainties further highlight the challenges clinicians face when assessing patients with suspected TBDs [34]. An Australian study investigated the variation between Australian laboratories performing serology for LB and reported that conflicting results existed between testing laboratories and that such discrepancies caused patients to question the validity of the results [35]. The authors aimed to find an agreement between assays commonly used for Bb s.l. serology tests. They concluded that there was discordance in the results between laboratories that was more likely due to operational and performance variations in assay testing algorithms. In the known seronegative population, the specificities of immunoassays ranged from 97.7–99.7%. In Australia, with its low prevalence of seropositivity, this would equate to a positive predictive value (PPV) of less than 4% [35]. The discrepancy in the laboratory results and low PPV aligns with recommendation 7 of the 2025 Senate inquiry, which urged the Australian Government to urgently review international diagnostic tests for tick-borne diseases and evaluate their suitability for domestic use [25]. The present study provides early evidence that a multiplex assay like TICKPLEX^®^ could be part of such a diagnostic expansion.

The aim of this proof-of-concept study was to evaluate the interlaboratory concordance of the TICKPLEX^®^ multiplex ELISA for detecting IgM and IgG antibodies to tick-borne pathogens using well-characterized Australian and international reference sera. This analysis provides preliminary evidence regarding the assay’s suitability for broader diagnostic application in Australia.

## 2. Materials and Methods

Within this study, two independent laboratories, Tezted Ltd. [Jyväskylä, Finland (FIN)] and Royal North Shore Hospital, New South Wales Pathology [North Sydney, Australia (AUS)], used the multiplex enzyme-linked immunosorbent assay TICKPLEX^®^ [36] to evaluate the performance of the test in Australia for the first time under Research Use Only (RUO) conditions using the optical density index (ODI). TICKPLEX^®^ has been validated by the manufacturer to determine TBD-specific antibody responses (IgG and IgM) in human sera or plasma with or without glycerol.

TICKPLEX^®^ PLUS (hereon referred to as index assay) is a CE-IVD registered product (European In-Vitro Diagnostic Devices Directive (98/79/EC) and legacy compliant 2017/746 IVDR) manufactured under ISO 13485:2016, the medical devices—quality management systems—requirements for regulatory purposes standard [https://www.iso.org/standard/59752.html (accessed on 8 December 2025)], accredited facility in Tezted Ltd., Jyväskylä, Finland. For this evaluation, we used the index assay RUO instructions (Version 17). This index assay was chosen because it can simultaneously test for IgM and IgG antibodies to multiple microbes [36]. If one were to replicate all the testing the index assay can do, 30 tests would be needed instead of 1. The index assay is an indirect ELISA that measures the immunoglobulin M (IgM) and immunoglobulin G (IgG) responses in human serum samples against *Borrelia burgdorferi* sensu lato species in spirochete and persistent forms, coinfections, and opportunistic microbes associated with a tick bite. The index assay follows standard indirect ELISA principles as described in the manufacturer’s instructions for use (IFU).

Specifically, the index assay includes antigens from *Borrelia burgdorferi sensu stricto*, *Borrelia afzelii*, *Borrelia garinii* in spirochete and persistent forms, *Babesia microti*, *Bartonella henselae*, *Ehrlichia chaffeensis*, *Rickettsia akari*, Coxsackievirus, Epstein–Barr virus, Human Parvovirus B19, *Mycoplasma fermentans*, and *Mycoplasma pneumoniae*. The spirochete form refers to the motile, spiral-shaped morphology of *Borrelia burgdorferi* typically seen during active infection [37,38,39]. ‘Persistent forms’ refer to alternative morphotypes such as round bodies or cyst-like structures that develop under stress conditions (e.g., immune pressure or antibiotic exposure) [37,38,40,41,42,43,44,45].

### 2.1. Material Used

Fifty-three serum samples were analyzed using the index assay to assess exposure to tick-borne disease-related pathogens. Of these samples, 33 were Lyme reference sera obtained from SeraCare Life Sciences Inc. (Milford, MA, USA) subsidiary in France and the Plasma Services Group (USA), while 20 were randomly selected well-characterized reference sera samples from Australia (Table 1). Inclusion criteria for the Australian samples consisted of well-characterized, de-identified serum from patients with suspected tick-borne illness, with or without prior Lyme-like disease (LLD) testing, and sufficient volume to allow for duplicate analysis. Exclusion criteria included samples with insufficient volume, compromised quality, or incomplete metadata regarding the clinical diagnosis. For the European and USA reference samples obtained from SeraCare Life Sciences Inc. and the Plasma Services Group, commercial sera were selected from predefined panels comprising Lyme-positive and Lyme-negative cases. These samples had been previously characterized using multiple FDA-cleared or CE-marked diagnostic assays.

To ensure unbiased analysis, the previous diagnoses of all patients were blinded for operators in both laboratories, and operators followed the instructions for use (IFU) for the index assay. Briefly, serum/plasma samples were stored at +4 °C for up to 7 days or at −20 °C for long-term storage prior to testing. For each assay, patient sera were freshly diluted in sample buffer at a 1:133 ratio for samples preserved in 30% glycerol and at a 1:200 ratio for undiluted samples, then pipetted into designated ELISA wells alongside negative, positive, and calibrator controls. Plates were incubated, washed, and processed sequentially with HRP-conjugated antibodies, substrate (TMB), and stop solution before absorbance was read at 450 nm. All samples were treated as potentially infectious, and handling precautions were strictly followed to prevent contamination and assay variation. The results from both laboratories were shared once the tests were completed.

To establish an inter-laboratory agreement, 28 out of the 53 serum samples had enough volume to be shared between Tezted Ltd. in Finland, and the Royal North Shore Hospital (New South Wales Health Pathology) in Sydney, Australia (Table 1). Among these shared samples, 22 of the 28, had been tested for Lyme like disease previously. The results were not known to the researchers, allowing for an unbiased assessment of agreement between the combined IgM and IgG index assay results in both sites. Out of the 53 reference samples, the remaining 25 were analyzed solely at Tezted Ltd. The IFU was adjusted accordingly by the manufacturer to account for glycerol dilution. The samples in 30% glycerol were further diluted at a 1:133 ratio in sample buffer, while undiluted sera samples were diluted at 1:200 (Appendix A).

The raw ODI values from the serum samples at 450 nm were normalized to determine the antibody response with cutoff ODI values for each antigen. Specifically, all pathogen negative, borderline, and positive responses were defined based on optical density values of lower than 0.90, within the range of 0.91 to 0.99, and exceeding 1.00, respectively. This normalization process ensured consistent interpretation of the test results across all samples, enabling comparison of immune responses to different tick-borne pathogens.

### 2.2. Internal and External Controls

The index assay includes internal positive (POS), negative (NEG), and coefficient of determination (R^2^) plate controls for both IgM and IgG. These controls are essential for verifying the accuracy of the test plate’s performance. The plate validity criteria for the IgM and IgG plate controls are as follows: a positive optical density (OD) reading should be above 1, while a negative OD reading should be below 0.5. Additionally, the R^2^ value should be greater than 0.75. An R-squared (R^2^), the coefficient of determination, was used to measure the interlaboratory variation between datasets presented by the two laboratories and how well a statistical model predicts a level of confidence in the results. In the case of paired data presented in this analysis, the R^2^ measures the proportion of variance shared by the two variables and ranges from 0 to 1. These criteria ensure the plate’s reliability and the validity of the test results.

This study incorporated external pre-characterized controls, consisting of reference serum samples obtained from the Australian laboratory, SeraCare Life Sciences Inc. (USA) subsidiary in France and the Plasma Services Group (USA). The Australian reference samples (*n* = 20) were randomly selected characterized undiluted serum samples from Australia (*n* = 17) which also included undiluted samples from the Plasma Services Group (PSG, *n* = 3) (Table 1). The Plasma Services Group (PSG) samples from the USA included symptomatic and asymptomatic individuals tested for LB. The PSG samples were provided with reference results from three different assays: the EUROIMMUN Lyme Western Blot for IgG and IgM, EUROIMMUN ELISA IgG and IgM, and Zeus Scientific ELISA IgG, IgM, and VLSE IgG/M. These assays target specific Lyme-related bands and antigens for accurate diagnosis. Similarly, serum samples from SeraCare Life Sciences Inc. were provided with reference results from eight different assays: DiaSorin LIAISON^®^ *Borrelia burgdorferi* Lyme IgM/IgG assay, Trinity Biotech Captia™ *Borrelia burgdorferi* IgG/IgM ELISA, BioMerieux VIDAS^®^ Lyme IgM II assay, MarDx *B. burgdorferi* Disease EIA IgM Test System, BioMerieux VIDAS^®^ Lyme IgG II assay, MarDx *B. burgdorferi* Disease EIA IgG Test System, MarDx *B. burgdorferi* (IgM) MarBlot Strip Test System, and MarDx *B. burgdorferi* (IgG) MarBlot Strip Test System.

The AccuSet™ Lyme Performance Panel 0845-0169 from SeraCare Life Sciences Inc. was also utilized. This panel comprised 15 undiluted plasma samples from multiple individuals who tested positive for Lyme borreliosis antibodies. Each sample represented a single collection event and provided a range of antibody reactivity for various Lyme IgM and IgG test methods. The panel included one nonreactive sample, a negative control, for all performed Lyme test methods. By including these external controls and panels of known samples, this study ensured a comprehensive evaluation of the index assay’s performance and ability to accurately detect antibodies associated with Lyme borreliosis.

### 2.3. Institutional Review Board Statement

Royal North Shore Hospital (Australia) offered de-identified, anonymized, and leftover human sera reference samples for this study for evaluation, and Tezted Ltd. did not have access to any private information (i.e., name, profession, or ethnicity) from the specimens that could be linked back to the patients. As such, informed consent was not collected, as the present study was not considered human subject research as mandated by the Declaration of Helsinki embodied in Common Rule set forth by the Code of Federal Regulations, USA [46,47] and the medical research act (Finland, 488/1999 (2.2.2001/101; section 20, 30.11.2012/689), which allows for the use of leftover and deidentified human serum samples without consent from the collection unit [Ministry 1 and 2]. In addition, characterized reference samples (*n* = 32) were purchased from Plasma Services Group (USA) and SeraCare Life Sciences Inc (Headquartered in the USA with a subsidiary in France).

### 2.4. Statistical Analysis

For quality control purposes, both laboratories, Finland, and Australia, conducted interplate and interoperator precision analyses by assessing the percentage of the coefficient of variance (CV%) or precision [standard deviation (SD) divided by the mean and multiplying by 100 to give CV%] on the optical density values for all plate controls and all microbial antigens [48]. The CV% was utilized to assess the performance and validity of the index assay for its intended purpose. The CV% indicates the percentage of variation within a dataset, with a minor variation indicating greater precision and better index assay performance. A lower CV% suggests that the index assay is suitable for accurately testing human IgM and IgG antibodies against various antigens associated with TBDs. To evaluate the CV% for the microbial antigens in the index assay, each operator repeatedly tested the negative serum control (TEZ1) provided in the kit in duplicate daily, with the Tezted laboratory performing the test 11 times and the Royal North Shore Hospital performing it 8 times. The CV% was calculated using the formula CV% = SD/mean. Generally, a CV% of less than 10 is considered very good, 10–20 is good, 20–30 is acceptable, and a CV% greater than 30 is unacceptable [48]. In this study, the calculated CV% for the index assay was less than or equal to 15%. Positive (PA), negative (NA), and overall (OA) agreements were calculated by comparing the index assay results with the final reference outcomes provided by the sample suppliers, which were determined according to the CDC two-tier criteria, as described by Watson and Petrie [49]. The reliability of each PA and NA was assessed using Cohen’s kappa (k) with a 95% confidence interval [49,50].

The kappa values were interpreted using standard thresholds for diagnostic agreement [50]. Cohen’s kappa (k) values are presented with a 95% confidence interval. Proportionate positive and negative agreements along with Cohen’s k were calculated using the EPITOOLS diagnostic test evaluation and comparison calculator, and the interrater reliability and proportional agreement analysis between various tests were carried out using only LB-positive and LB-negative patient groups [https://epitools.ausvet.com.au/comparetwotests (accessed on 30 August 2022)]. Fisher’s exact test assessed the significant differences in IgM or IgG immune responses between LB (positive and negative) and the Australian cohort. The two-tailed *p* values for Fisher’s exact test were calculated using the 2 × 2 contingency table on GraphPad [https://www.graphpad.com/quickcalcs/contingency1/ (accessed on 8 December 2025)], and Fisher’s exact test results with *p* values < 0.05 were considered statistically associated [51].

## 3. Results

Initially, performance validity criteria according to the manufacturer’s quality control standard were assessed to ensure that the index assay was performed correctly and used as intended at both independent laboratories in Finland and Australia (Appendix A). All IgM and IgG internal plate controls (i.e., POS, NEG, and R^2^) passed the plate validity criteria demanded by the manufacturer (Appendix A). In addition, we measured precision (CV%) to be less than 15% for the POS and R^2^ internal controls, demonstrating low interplate variability at both independent laboratory sites (Appendix A). In addition, the CV% for IgM and IgG internal negative control datasets between the two labs were acceptable and consistent with a CV% of greater than 15% because the optical density values approach the lower limit of quantification. According to the CV% analysis in Appendix A, the two sites had consistent results when compared, suggesting that the index assay can be used regularly.

Additionally, for quality control, interplate and interoperator precision analyses were conducted (Figure 1). The overall agreement percentages ranged between 68–82% and 86–100% for IgM and IgG, respectively. The Cohen’s kappa (k) values ranged from 0.24 (fair agreement) to 0.61 (substantial agreement) for IgM and 0.58 to 0.60 (moderate agreement overall). The k value is not applicable (N/A) for an antigen in the index assay when all specimens in Appendix A produced only negative or positive responses in Finland and Australia.

The lowest IgM interlaboratory agreement from Figure 1 indicated that Coxsackievirus had 68% interlaboratory agreement, k = 0.27 (95% CI 0.00–0.54), and Mycoplasma fermentans and *Mycoplasma pneumoniae* had 68% interlaboratory agreement, k = 0.24 (95% CI = −0.13–0.62). The highest IgG interlaboratory agreement from Figure 1 shows that Epstein–Barr virus had 89% interlaboratory agreement, k = 0.60 (95% CI 0.19–1.00), and Borrelia persistent forms had 93% interlaboratory agreement, k = 0.78 (95% CI = 0.51–1.06). The overall agreement (%) for IgG and IgM, as shown in Figure 1, was calculated from the mean ODI (optical density index) from each laboratory (AUS and FIN) as per Appendix A and compared for interlaboratory agreement of each analyte per specimen. Taken together, these findings indicate moderate reproducibility for IgM and stronger, substantial agreement for IgG, suggesting that IgG responses may offer more consistent interlaboratory performance within this multiplex platform.

IgM, IgG, and IgM/IgG combined positive agreement were 97%, 53%, and 88%, respectively (Figure 2). The negative agreement (%) for IgM, IgG, and IgM/IgG combined were 93%, 79%, and 67%, respectively. The overall agreement for IgM, IgG, and IgM/IgG combined was 95%, 68%, and 81%, respectively. Cohen’s kappa (k) value was almost perfect agreement for IgM (k = 0.89), fair agreement for IgG (k = 0.30), and moderate agreement for IgM/IgG combined (k = 0.54). Laboratories tested the samples and referenced Lyme borreliosis test results (*n* = 22). The clinical outcomes were determined using the CDC two-tier criteria [52], as shown in Appendix A, for the reference diagnosis of each specimen. The agreement between the mean ODI of IgM and IgG was analyzed using data from both laboratories (FIN and AUS). Appendix A displays the specimen wise analysis, while Appendix A shows the mean ODI values for IgM and IgG. Specimen IDs AU6 and AU31 are the same specimens, but with one difference: AU6 was stored in 30% glycerol before testing, while AU31 was not stored in glycerol at −20 °C. Despite this difference, both specimens had similar clinical outcomes, as demonstrated in Appendix A. It is worth noting that AU31 was excluded from clinical assessments during the analysis between the index and reference tests, as shown in Figure 2.

Unlike Borrelia spirochete species, the IgM and IgG percentages of positive immune responses for the Borrelia persistent forms were higher in Australian reference samples compared to the other shared reference specimens (Figure 3). Moreover, the proportion of IgM and IgG responses in the Australian group was frequently more elevated than that in the reference sera for tick-borne coinfections (text in orange) and related opportunistic microbes (text in green) (Figure 3). In general, the IgM responses had a higher percentage of positivity, ranging from 35% to 70%. These responses were found to be higher in Australian samples (*n* = 20) compared to the reference specimens (*n* = 32). The proportion of immune responses against all antigens on the index assay was determined by the mean ODI of both laboratories (AUS and FIN) from Appendix A for IgM and Appendix A for IgG.

In addition, we evaluated the possibility that LLD is associated with coinfections with tickborne pathogens and other pathogens that may or may not be tickborne. Figure 4 shows that the Lyme reference samples, and the Australian samples produced IgM and IgG antibodies to multiple microbes, some potentially associated with a tick bite. The association of antibodies to multiple microbes as well as *Borrelia burgdorferi* was common even among the Australian samples (Figure 4).

## 4. Discussion

This investigation focused on the performance and reliability of a multiplex enzyme-linked immunosorbent assay (TICKPLEX^®^) using the ODI, in detecting tick-borne diseases by comparing results obtained from two distinct institutions, Tezted Ltd. in Finland and the Royal North Shore Hospital in Australia. This was achieved by using defined blindly tested reference samples to minimize bias between two laboratories. We tested and evaluated patients’ immune responses blindly and then compared these findings to established reference values available from the Australian laboratory, SeraCare Life Sciences Inc. and the Plasma Services Group.

The index assay was selected for its ability to measure both IgM and IgG antibodies that target Borrelia’s pleomorphic forms [37], including the round body or persistent forms, as well as a range of common opportunistic pathogens and co-infections simultaneously. The scientific community is reporting the significance of Borrelia’s pleomorphic forms, like the findings by Golovchenko et al., who identified cyst-like structures of Borrelia in a patient’s brain tissue with confirmed Lyme borreliosis diagnosis and ongoing symptoms despite undergoing treatment [53].

Moreover, the literature documents the prevalence and clinical implications of co-infections, such as those caused by *Babesia* spp. [54], *Bartonella* spp. [55], *Ehrlichia* spp. [56], and *Rickettsia* spp. [57,58]. Concurrently, there is an escalating awareness regarding the differential diagnostic process for patients with conditions that overlap symptomatically with tick-borne diseases, especially chronic fatigue syndrome [59]. In these cases, opportunistic pathogens, including Coxsackievirus [60,61], Epstein–Barr virus [62,63,64], Human Parvovirus B19 [65,66], and *Mycoplasma* spp. [67,68,69,70], are increasingly recognized for their potential to exacerbate other medical conditions, particularly in individuals with compromised immune systems, underscoring the multifaceted nature of TBD diagnosis and management.

### 4.1. Assay Performance and Interlaboratory Concordance

The performance of the index assay showed substantial interlaboratory agreement (Figure 1) for IgG, 86% to 100%, and moderate agreement for IgM, 68% to 82%, for samples tested in both Tezted Ltd. and Royal North Shore Hospital laboratories in Finland and Australia, respectively (*n* = 28). Overall, the interlaboratory agreement percentage between the Royal North Shore Hospital and Tezted Ltd. laboratories (*n* = 28) was relatively higher for IgG, 86% to 100%, than IgM, 68% to 82%. As demonstrated by Cohen’s kappa (k), the inter-reliability for IgM ranges from fair to substantial agreement (k = 0.24 to 0.61) when compared to IgG ranging from moderate to substantial agreement (k = 0.58 to 0.78). The IgM interlaboratory clinical agreement for Borrelia spirochete species shows only a fair agreement (k = 0.38). For an IgM result, as shown in Appendix A, the precision for both laboratories ranged from 0% to 3.65% for positive control values, demonstrating that the precision of techniques for both laboratories is not the cause of lower fair agreement. Borrelia serological tests have a higher potential for false positive results when testing for LB [71]. Therefore, the two-tier CDC criteria (CDC 2019) were used to calculate clinical agreement in both reference sera and Australian samples, as shown in Appendix A.

The combined IgM and IgG clinical agreement for Lyme borreliosis between the index assay and reference test results was satisfactory (Figure 2). The samples that were tested by both laboratories and reference Lyme borreliosis test results (*n* = 22) were interpreted using the CDC two-tier criteria [72] and showed a positive agreement value of 97% for IgM, 53% for IgG, and 88% agreement for IgM and IgG combined (Figure 2).

The lower IgG clinical agreement in Figure 2 is due to the pre-characterization of the sera using either EUROIMMUN or *Zeus borrelia* spp. IgG and IgM assays. Therefore, the comparison for the clinical agreement was calculated with IgM and IgG combined [73,74,75,76,77]. This, however, was not reflected in the negative agreement (%) for IgM, IgG, and IgM/IgG, as most reference sera, *n* = 22, were known as positive samples for *Borrelia* spp. Some of the reference sera had been only characterized by EUROIMMUN ELISAs that contain antigens of *Borrelia burgdorferi*, *Borrelia afzelii*, *Borrelia garinii*, and Borrelia VlsE. Some sera were characterized with Zeus EIA assays with B. burgdorferi only as an antigen. In contrast, some Australian samples were only characterized by DiaSorin Liaison XL B. burgdorferi IgG and IgM assays or the Trinity Biotech EU Lyme Western blot. The lower IgG clinical agreement in Figure 2 likely reflects differences in antigenic composition between reference assays, and may also indicate variability in regional strain diversity or in the maturation of IgG responses across cohorts [78,79].

### 4.2. Interpretation of Serological Findings

In our analysis presented in Figure 3, we observed that the antibody responses for both IgM and IgG to co-infections and opportunistic pathogens were notably higher in Australian reference specimens compared to reference specimens from the USA. Expressly, while the IgG-positive response rates showed a similar range for both reference (0% to 19%) and Australian shared samples (5% to 15%), a marked difference was noted in IgM-positive response rates, with the reference samples showing 16% to 44% and Australian samples demonstrating 35% to 65%. Among these comparisons, only those with Fisher’s exact test *p* < 0.05 met the threshold for statistical significance, indicating that certain antigen-specific IgM differences between cohorts were statistically associated, whereas most IgG differences were not. These findings support the Australian Senate concerns that current diagnostic tests may be missing significant immune responses to other tick-borne microbes beyond *Borrelia* spp. The Senate emphasized that the limitations in current testing may contribute to ongoing misdiagnosis or underdiagnosis of TBDs in Australia [25].

Furthermore, as illustrated in Figure 4, the co-infection rates with Borrelia for IgM and IgG were 53% and 13% in the reference group, contrasting with 65% and 25%, respectively, in the Australian reference samples. This discrepancy underscores a significant variation compared to co-infection rates reported between 0.7% and 41.6% in European and U.S. studies [80]. Moreover, Ixodes tick species in Australia can transmit *Coxiella burnetii*, *Rickettsia australis*, *Rickettsia honei*, and *Rickettsia honei* subspecies marmioni. Australian Ixodid ticks could transmit *Babesia microti*, *Anaplasma* spp. *Bartonella* spp. *Burkholderia* spp., *Francisella* spp., and even viruses, as listed in an Australian TBD review [81].

We aim to highlight differences in the co-infection rates not to assert our findings in Figure 3 and Figure 4 as conclusive but to emphasize the complexity of the tick-borne disease landscape in Australia, which appears to be more complex than previously recognized, thereby meriting deeper investigation. We acknowledge the inherent challenges of serological testing for Lyme and related TBDs, especially concerning cross-reactivity [82,83], variable immune response, timing of serum sampling following exposure to tick-borne pathogens [84], and the inherent issues with tests for IgM antibodies [85,86] introduce complexities in accurately interpreting seroprevalence and co-infection data. Thus, these nuances demand further research to ensure a robust understanding of the findings.

It is also important to note that 5 of 28 samples, as seen in Appendix A, tested positive for IgM in all markers on the index assay. Since the sera were from patients with a clinical diagnosis of LB, as confirmed by other studies and tests, this illustrates that *Borrelia* spp. serology assays, like all serology assays, have the potential to yield false positive IgM results in patients who have active viral infections [87] due to polyclonal activation of T and B lymphocytes [82]. Similarly, Borrelia IgM antibody persistence after treatment has also been demonstrated in various studies [71,85,86,88,89,90], and this limits its use in diagnosis.

### 4.3. Study Limitations and Future Directions

Finally, the relatively small number of specimens limits statistical power, particularly for interlaboratory comparisons. Nonetheless, these results provide proof-of-concept evidence supporting the feasibility of multiplex serological testing for Australian TBDs. Future work should include multicenter validation with larger sample sets and the integration of molecular diagnostics such as PCR or antigen detection to improve clinical interpretation.

## 5. Conclusions

The purpose of this study was to compare the results obtained by two separate laboratories using the TICKPLEX^®^ (index assay) on the same serum samples for detection of antibodies to known and potential agents infecting patients with TBD. High concordance was demonstrated between the two laboratories for IgG and IgM antibody responses in reference samples to multiple microbial antigens of tick-borne disease pathogens, as expected when compared to the referenced results. The clinical correlation, however, of Australian sera (*n* = 20) for microbes implicated in tickborne coinfections and other opportunistic infections was challenging, as shown in Appendix A, due to a small sample size (*n* = 20). Tests available in Australia for TBD were limited to antibody tests for *Rickettsia australis* and *Borrelia burgdorferi* s.l. Whilst there was likely concordance between those tests and their respective index assay tests-*Rickettsia akari* (for *R. australis*) and *Borrelia* spp. [*Borrelia afzelii*, *Borrelia burgdorferi*, *Borrelia garinii* for both spirochete and persistent forms] (for *B. burgdorferi* s.l.), the antibodies sought were not from the same species. This may compromise the ability to confirm the diagnosis of Rickettsial spotted fever and Lyme borreliosis in Australian patients, but this could also be investigated by using a larger number of samples. Similarly, the reference test results for *Bartonella henselae* were for total antibody results, while, in the index assay, IgG and IgM were tested separately.

There is a concern in Australia from patients suffering from symptoms of chronic debilitating illness which they temporally relate to tick bite but for which no definitive cause is found. This leaves patients confused, suffering, and despondent, and they are at risk of receiving incorrect or inappropriate diagnoses and treatments. The severity of illness varies among patients and can often overlap with other underlying multisystem disorders, such as chronic fatigue syndrome/myalgic encephalomyelitis (CFS/ME). Limited diagnostic tests are available in Australia, which only exacerbates the situation. Our findings provide timely data to support several recommendations of the 2025 Australian Senate report, which called for immediate investment in broader TBD diagnostic research, stakeholder-engaged clinical pathways, and the evaluation of internationally developed tests for use in the Australian context [25]. The index assay, TICKPLEX^®^, may offer one such tool, meriting further study across larger Australian cohorts.

## Figures and Tables

**Figure 1 pathogens-14-01281-f001:**
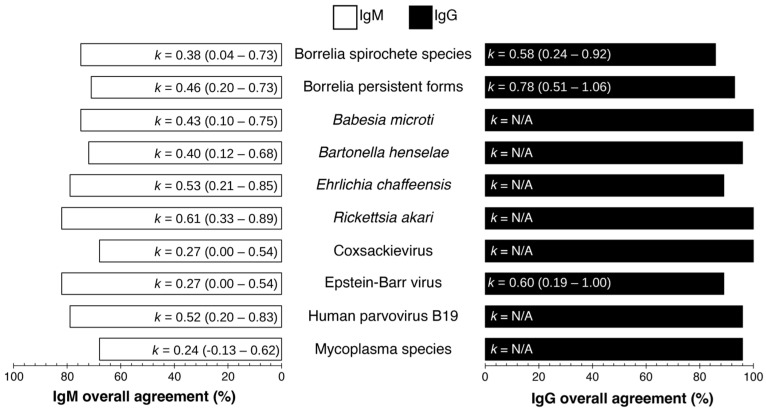
The overall interlaboratory clinical agreement for the index assay between different laboratories was moderate for IgM and substantial for IgG. Appendix A (IgM) and Appendix A (IgG) demonstrate a list of common specimens (*n* = 28) obtained in Finland and Australia with their clinical outcomes used for calculating the interlaboratory clinical agreement. In Finland and Australia, when all specimens in Appendix A consistently generated only negative or positive responses, the k value for an antigen in the index assay becomes not applicable (N/A). Borrelia spirochete species and persistent forms refer to *Borrelia burgdorferi sensu stricto*, *Borrelia afzelii*, and *Borrelia garinii* in spirochete and persistent forms. Similarly, Mycoplasma species refers to *Mycoplasma fermentans* and *Mycoplasma pneumoniae*.

**Figure 2 pathogens-14-01281-f002:**
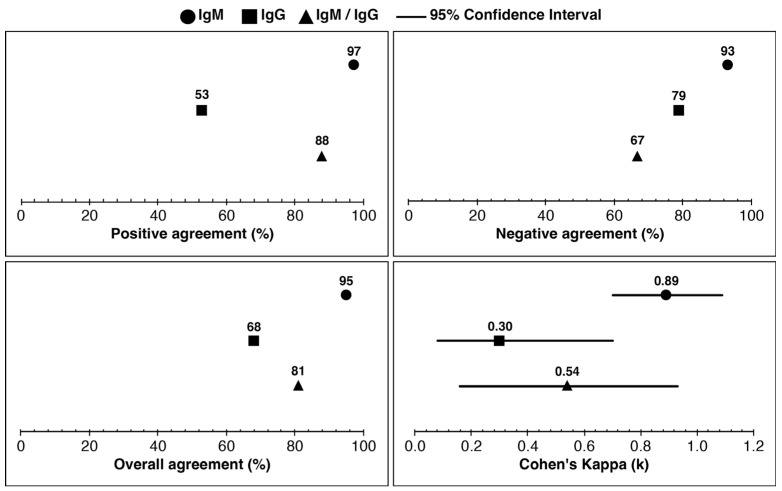
The combined IgM and IgG agreement for Lyme borreliosis between the index and reference test results is moderate. Appendix A (IgM) and Appendix A (IgG) demonstrate a list of common specimens performed in Finland and Australia, with their clinical outcomes used to calculate clinical agreement with previous diagnostic findings for samples in Appendix A (*n* = 22). Reference Lyme borreliosis test results were interpreted for IgM and IgG following the CDC two-tier criteria (CDC 2019) and are presented in Appendix A. An IgM- or IgG-positive reference test for Lyme borreliosis was interpreted as overall positive for IgM and IgG combined analysis (Appendix A). For the index assay, the final interpretation of the result for IgM, IgG, or IgM and IgG applied the most prevalent clinical result between the labs for a specimen. For example, a sample with three out of four negative IgM or IgG immune responses to Borrelia antigens on the index assay was interpreted as negative. If the sample presented two positive and two negative IgM or IgG clinical findings on the index assay, the specimen was considered positive, following the same standard applied to the reference test results in Appendix A.

**Figure 3 pathogens-14-01281-f003:**
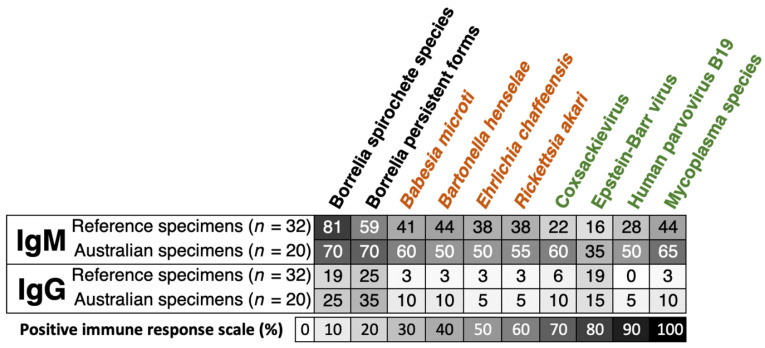
Screening Australian patients for tick-borne disease-related microbes could be a missed opportunity to reduce misdiagnosed and undiagnosed patient cases. The proportion of positive IgM and IgG immune responses for TICKPLEX^®^ antigens was comparable between the Australian (*n* = 20) and Lyme reference (*n* = 32) specimens. Borrelia spirochete species and persistent forms refer to *Borrelia burgdorferi sensu stricto*, *Borrelia afzelii*, and *Borrelia garinii* in spirochete and persistent forms. Similarly, Mycoplasma species refers to *Mycoplasma fermentans* and *Mycoplasma pneumoniae*. The Borrelia antigens are labelled in black, co-infections in orange, and opportunistic pathogens in green. The varying shades from white to black represent percentage positivity, where increased darkness indicates a higher degree of positivity. Appendix A present clinical outcomes for all samples used in Figure 3.

**Figure 4 pathogens-14-01281-f004:**
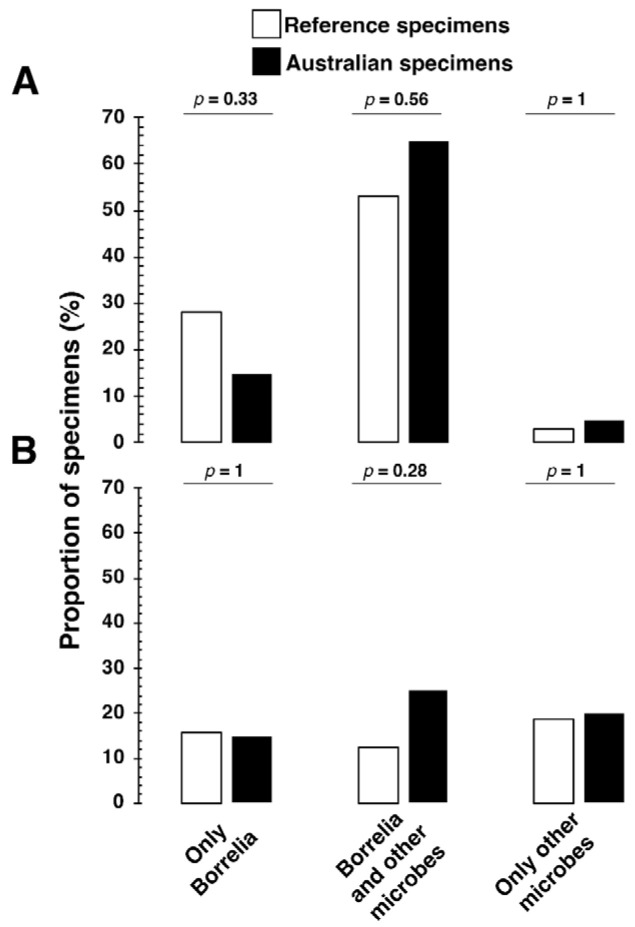
The Lyme reference and Australian reference specimens produced (**A**) IgM and (**B**) IgG responses to multiple microbial antigens associated with tick-borne diseases. The proportion of positive IgM and IgG immune responses to Borrelia and other microbes was higher in Australian shared samples (*n* = 20) than in Lyme reference samples (*n* = 32). Appendix A present clinical outcomes for all specimens used in Figure 4. Only Borrelia refers to Borrelia spirochete species and persistent forms of Borrelia (*Borrelia burgdorferi sensu stricto*, *Borrelia afzelii*, and *Borrelia garinii*). Likewise, other microbes refer to coinfections (*Babesia microti*, *Bartonella henselae*, *Ehrlichia chaffeensis*, *Rickettsia akari*) and opportunistic microbes (*Coxsackievirus*, *Epstein–Barr virus*, *human parvovirus B19*, *Mycoplasma fermentans*, and *Mycoplasma pneumoniae*) in the index assay. Fisher’s exact test assessed the differences in IgM and IgG immune responses between the Australian (*n* = 20) and Lyme reference samples (*n* = 32). A two-tailed *p*-value for Fisher’s exact below 0.05 was considered statistically associated.

**Table 1 pathogens-14-01281-t001:** Summary of sample origin, total sample size, and shared specimens used for interlaboratory comparison.

Total Number of Samples (*n*)	53
Reference serum samples from SeraCare Life Sciences Inc. and Plasma Services Group (*n*)	33 [Includes two samples labeled as AU6 and AU31 (Appendix A), which are identical. These samples were tested in the presence and absence of 30% Glycerol]
Reference serum samples from Australia (*n*)	20
Common serum samples shared between the independent labs from the total number of samples (*n*)	28 (22 samples had tested seropositive for IgM and/or IgG or both for Lyme borreliosis)

## Data Availability

All data supporting the reported results are included in this article and its Appendix A. No additional datasets were generated or analyzed during this study.

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
