# Peer review of "Interlaboratory Concordance of a Multiplex ELISA for Lyme and Lyme-like Illness Using Australian Samples and Commercial Reference Panels: A Proof-of-Concept Study"

_pathogens, 2025, doi:10.3390/pathogens14121281_

Round 1
Reviewer 1 Report (Previous Reviewer 3)
Comments and Suggestions for Authors
Garg K. et al: Interlaboratory Concordance of a Multiplex ELISA for Lyme and Lyme-like Illness: A Proof-of-Concept Study Using Australian Samples and U.S.A./European Lyme Reference Sera [resubmission V.1; in toto V.2]
This referee finds (1) the new version of the manuscript benefiting from changes the authors made in response to criticism of their earlier text (initially titled: Assessing interlaboratory performance and concordance of tick-borne disease testing using multiplex enzyme-linked immunosorbent assay), and (2) those points that hasn't been reflected in the revision at least adequately addressed in the authors covering letter. As explained in that letter, the Australian cooperating staff at the Royal North Shore Hospital prefers not to co-author this article, which lefts its main weakness - the circumstance it is authored by employees/shareholders of a company manufacturing the diagnostic kit in question - unreinforced and liable to speculations over an unexcludible bias (whatever this referee didn't recognized signs of non-objectivity).
Anyway, this referee can corroborate that Lyme borreliosis-mimicking symptoms present a neglected yet sporadically encountered diagnostic problem even beyond Australia in LB-endemic areas - e.g. rarely developing in patients positively bitten by insects, not ticks - which hasn't so far been satisfactorily answered, and advocates therefore for publication of this matter as a contribution to filling that gap.
Minor issues:
P1,L4: full stop in U.S.A. (USA ?)
P1,L33 : like a name of any other infection ending in '-osis', "borreliosis" is commonly typeset in lower case letters - check it throughout the manuscript, pls
P7, L286: delete "or dependent", pls. ('dependence' implies causality which no statistical test can prove..)
P15,L650+4: "CV" should read "CV%" as defined on p.6,l.256
P16-23: the list of references seems to be overly extensive (ca. one third of the article's space is occupied by references!) - approaching some state-of-the-art reviews - the authors may consider its thinning ..
P17,L694: check the author or indicate as "Anonymous", pls
P18,L723: lowercase the names of authors, pls.
P19,L750: ditto
Suppl. Tab.2: Pls., insert spaces or thicken the vertical lines separating the POS/NEG/R2 triplets of columns to be easily distinguished on both pages; the bottom CV% in the first column is just a fill and could be omitted..
Suppl. Tabs 8 and 9: Pls., in order to make it explicit and avoid confusion of the reader, explain in the header what the "Agreement?" classification means, and either delete the two void rows (N/As) or (better?) append one to make the Tabs consistent..
Comments on the Quality of English Language
this referee isn't a native English speaker
Author Response
Reviewer 1 comment 1
P1,L4: full stop in U.S.A. (USA ?)
Author response 1
Corrected to USA in line 257.
Reviewer 1 comment 2
P1,L33 : like a name of any other infection ending in '-osis', "borreliosis" is commonly typeset in lower case letters - check it throughout the manuscript, pls
Author response 2
Lyme borreliosis has been applied throughout the manuscript.
Reviewer 1 comment 3
P7, L286: delete "or dependent", pls. ('dependence' implies causality which no statistical test can prove..)
Author response 3
Removed “or dependent” when describing the use of Fisher’s exact test throughout the manuscript.
Reviewer 1 comment 4
P15,L650+4: "CV" should read "CV%" as defined on p.6,l.256
Author response 4
Corrected in the abbreviations section and throughout the manuscript.
Reviewer 1 comment 5
P16-23: the list of references seems to be overly extensive (ca. one third of the article's space is occupied by references!) - approaching some state-of-the-art reviews - the authors may consider its thinning
Author response 5
We appreciate the comment; however, given the emerging nature of this field, we believe it is important to retain the current references to ensure statements are well supported. Many cited works are Australian tick-borne disease studies that are underrepresented in the literature, and their inclusion helps provide necessary context for our findings.
Reviewer 1 comment 6
P17,L694: check the author or indicate as "Anonymous", pls
Author response 6
Author in reference 25 corrected.
Reviewer 1 comment 7
P18,L723: lowercase the names of authors, pls.
Author response 7
Reference 30 corrected.
Reviewer 1 comment 8
P19,L750: ditto
Author response 8
Reference 41 corrected.
Reviewer 1 comment 9
Suppl. Tab.2: Pls., insert spaces or thicken the vertical lines separating the POS/NEG/R2 triplets of columns to be easily distinguished on both pages; the bottom CV% in the first column is just a fill and could be omitted
Author response 9
Dashed vertical lines have been added in Table S2 to separate POS/NEG/R2.
Reviewer 1 comment 10
Suppl. Tabs 8 and 9: Pls., in order to make it explicit and avoid confusion of the reader, explain in the header what the "Agreement?" classification means, and either delete the two void rows (N/As) or (better?) append one to make the Tabs consistent.
Author response 10
Agreement has been explained in Table S8 and S9 headers.
Reviewer 2 Report (New Reviewer)
Comments and Suggestions for Authors
The study by Garg et al provides a proof-of-concept evaluation of the TICKPLEX® multiplex ELISA for detecting antibodies to multiple tick-borne disease (TBD) agents across northern and southern hemisphere samples. Two independent laboratories analyzed a panel of 53 sera, including 33 confirmed northern hemisphere TBD cases and 20 Australian samples from patients with suspected Lyme-like disease. The assay demonstrated high inter-laboratory concordance for IgG and IgM detection, indicating technical robustness and reproducibility. However, interpretation of the Australian data was limited by the small sample size and antigenic differences between TICKPLEX® targets and locally available diagnostic tests. The findings are preliminary but support further large-scale validation of TICKPLEX® for its potential utility in expanding TBD diagnostic capabilities in Australia, where Lyme borreliosis remains unproven and reliable serological tools are lacking.
Minor comments:
- lines 35–41-Much of the epidemiological data could be condensed. Repetition of citations [2] and the extensive numeric listing of regional seroprevalence distracts from the main narrative.
- The discussion of Australian Senate reports, while important, occupies a large portion of the introduction and overshadows the scientific rationale. Consider summarizing policy content more succinctly and linking it explicitly to the diagnostic need.
- Lines 91–101- provide an exhaustive description of diagnostic methods (PCR, IFA, ELISA, WB). While accurate, this could be shortened, especially as these are well-established methods.
- Paragraph transitions are sometimes abrupt (e.g., between the Senate inquiry section and endemic TBDs). Adding brief transition sentences would help maintain logical continuity.
- Lines 117–121- The study’s objective appears somewhat buried. It should be clearly delineated preferably as a short, standalone paragraph at the end of the introduction to emphasize the purpose and novelty of the current work.
- Lines 141–146 repeat generic ELISA details that could be summarized in a single sentence (“The assay follows standard indirect ELISA principles as described in the manufacturer’s IFU”).
- The text occasionally revisits the same methodological details (e.g., glycerol dilution adjustments, source of control sera) in multiple places. Consolidating related information under concise subsections would improve flow.
- Lines 181–187- The description of sample distribution across sites is complex. A schematic or summary table showing sample origin, number tested per lab, and overlap would enhance clarity.
- Ensure consistent use of “Index Assay,” “TICKPLEX®,” and “TICKPLEX® PLUS” to prevent confusion.
- When describing “persistent forms,” it may help to use standardized terminology (“round bodies” or “cyst-like morphotypes”) and remove speculative phrasing (e.g., “hypothesized to play a role”).
- While comprehensive, it could be streamlined by removing textbook definitions (e.g.,kappa ranges) and referencing standard interpretive thresholds instead.
- Several details (e.g., CV% explanations, replication of figure legends within text) are repeated and could be summarized more succinctly. This would enhance clarity without compromising rigor.
- The section reports many kappa values but does not interpret their diagnostic implications. A brief synthesis (e.g., “moderate IgM reproducibility but weaker IgG concordance”) would help readers grasp the overall assay performance.
- While the Australian versus reference cohort findings (Figures 3–4) are intriguing, the results lack clear statistical summaries (e.g., specific p-values or confidence intervals). Clarifying which associations reached significance would strengthen interpretability.
- In discussion some content reiterates numerical results (e.g., % agreement, kappa values, and assay details) already presented in the Results. These should be summarized conceptually, e.g., “moderate IgM reproducibility and substantial IgG consistency” rather than restated in detail.
- Lines 418–426- The paragraphs on opportunistic pathogens and vector ecology (lines 474–477) provide valuable background but would fit better in the Introduction or could be condensed to maintain focus on the study’s data-driven insights.
- The discussion intermingles assay performance evaluation with epidemiological speculation. It would benefit from clearer delineation between (i) assay validation, (ii) interpretation of serological findings, and (iii) implications for Australian TBD surveillance.
- The section acknowledges variable agreement but could more explicitly interpret what this means diagnostically e.g., whether the moderate IgG agreement suggests antigenic differences, reference assay variability, or potential regional strain diversity.
- The final paragraph appropriately notes the small sample size but could be strengthened by a succinct summary statement emphasizing the study’s proof-of-concept contribution and the specific next steps (e.g., multicenter validation, inclusion of PCR or antigen detection).
Author Response
Reviewer 2 comment 1
lines 35–41-Much of the epidemiological data could be condensed. Repetition of citations [2] and the extensive numeric listing of regional seroprevalence distracts from the main narrative.
Author response 1
Condensed in lines 35 to 39.
Reviewer 2 comment 2
The discussion of Australian Senate reports, while important, occupies a large portion of the introduction and overshadows the scientific rationale. Consider summarizing policy content more succinctly and linking it explicitly to the diagnostic need.
Author response 2
Summarized in lines 73 to 78.
Reviewer 2 comment 3
Lines 91–101- provide an exhaustive description of diagnostic methods (PCR, IFA, ELISA, WB). While accurate, this could be shortened, especially as these are well-established methods.
Author response 3
Shortened in lines 97 to 101.
Reviewer 2 comment 4
Paragraph transitions are sometimes abrupt (e.g., between the Senate inquiry section and endemic TBDs). Adding brief transition sentences would help maintain logical continuity.
Author response 4
Brief transition added in lines 101 to 105.
Reviewer 2 comment 5
Lines 117–121- The study’s objective appears somewhat buried. It should be clearly delineated preferably as a short, standalone paragraph at the end of the introduction to emphasize the purpose and novelty of the current work.
Author response 5
Study objectives clarified in lines 164 to 168.
Reviewer 2 comment 6
Lines 141–146 repeat generic ELISA details that could be summarized in a single sentence (“The assay follows standard indirect ELISA principles as described in the manufacturer’s IFU”).
Author response 6
Generic ELISA details removed from said lines and replaced with a single line from 187 to 188.
Reviewer 2 comment 7
The text occasionally revisits the same methodological details (e.g., glycerol dilution adjustments, source of control sera) in multiple places. Consolidating related information under concise subsections would improve flow.
Author response 7
We have removed redundant information where possible while preserving the clarity and context of the manuscript.
Reviewer 2 comment 8
Lines 181–187- The description of sample distribution across sites is complex. A schematic or summary table showing sample origin, number tested per lab, and overlap would enhance clarity.
Author response 8
Table 1 highlights the 28 shared samples analyzed by both sites, and we have revised the table legend to explicitly indicate this purpose.
Reviewer 2 comment 9
Ensure consistent use of “Index Assay,” “TICKPLEX®,” and “TICKPLEX® PLUS” to prevent confusion.
Author response 9
The use of “index assay” has been made consistent in the manuscript.
Reviewer 2 comment 10
When describing “persistent forms,” it may help to use standardized terminology (“round bodies” or “cyst-like morphotypes”) and remove speculative phrasing (e.g., “hypothesized to play a role”).
Author response 10
We have modified lines 194 to 196.
Reviewer 2 comment 11
While comprehensive, it could be streamlined by removing textbook definitions (e.g.,kappa ranges) and referencing standard interpretive thresholds instead.
Author response 11
Streamlined in lines 374 to 375.
Reviewer 2 comment 12
Several details (e.g., CV% explanations, replication of figure legends within text) are repeated and could be summarized more succinctly. This would enhance clarity without compromising rigor.
Author response 12
Repeating elements in supplementary table legends have been removed.
Reviewer 2 comment 13
The section reports many kappa values but does not interpret their diagnostic implications. A brief synthesis (e.g., “moderate IgM reproducibility but weaker IgG concordance”) would help readers grasp the overall assay performance.
Author response 13
A brief synthesis has been added in lines 431 to 434.
Reviewer 2 comment 14
While the Australian versus reference cohort findings (Figures 3–4) are intriguing, the results lack clear statistical summaries (e.g., specific p-values or confidence intervals). Clarifying which associations reached significance would strengthen interpretability.
Author response 14
Clarification has been added in lines 594 to 597.
Reviewer 2 comment 15
In discussion some content reiterates numerical results (e.g., % agreement, kappa values, and assay details) already presented in the Results. These should be summarized conceptually, e.g., “moderate IgM reproducibility and substantial IgG consistency” rather than restated in detail.
Author response 15
The discussion section presents conceptual summaries such as ‘substantial agreement’ or ‘fair agreement,’ accompanied by the corresponding statistics in brackets. We have retained both to ensure consistency in how results are reported throughout the manuscript.
Reviewer 2 comment 16
Lines 418–426- The paragraphs on opportunistic pathogens and vector ecology (lines 474–477) provide valuable background but would fit better in the Introduction or could be condensed to maintain focus on the study’s data-driven insights.
Author response 16
We appreciate the reviewer’s suggestion. However, the paragraphs referenced (lines 418–426 and 474–477) are included in the Discussion specifically to contextualize the immune response patterns observed in Figures 3–4. The differences between Australian and international samples require ecological and pathogen-related interpretation, which would not fit naturally in the Introduction where results are not yet presented.
Reviewer 2 comment 17
The discussion intermingles assay performance evaluation with epidemiological speculation. It would benefit from clearer delineation between (i) assay validation, (ii) interpretation of serological findings, and (iii) implications for Australian TBD surveillance.
Author response 17
To improve clarity and thematic separation, we have reorganized the Discussion into distinct subsections addressing (i) assay performance and interlaboratory validation, (ii) interpretation of serological findings, and (iii) implications for Australian tick-borne disease surveillance.
Reviewer 2 comment 18
The section acknowledges variable agreement but could more explicitly interpret what this means diagnostically e.g., whether the moderate IgG agreement suggests antigenic differences, reference assay variability, or potential regional strain diversity.
Author response 18
Explicit interpretation added in lines 587 to 590.
Reviewer 2 comment 19
The final paragraph appropriately notes the small sample size but could be strengthened by a succinct summary statement emphasizing the study’s proof-of-concept contribution and the specific next steps (e.g., multicenter validation, inclusion of PCR or antigen detection).
Author response 19
The final paragraph in the discussion section has been strengthened in lines 637 to 642.
This manuscript is a resubmission of an earlier submission. The following is a list of the peer review reports and author responses from that submission.
Round 1
Reviewer 1 Report
Comments and Suggestions for Authors
TBDs pose a major challenge to modern health services. More and more ticks are carrying pathogens dangerous to humans. The small size of these arachnids and the delayed onset of symptoms mean that it is often difficult to determine the cause of non-specific symptoms without a detailed laboratory diagnosis, and without a clear diagnosis, treatment is very difficult. Therefore, the creation of a test to determine if and which tick-borne disease is present is extremely valuable. Below is a list of considerations:
Line 13 - the name borreliosis should also be inserted, similarly line 30
Line 61 - in the introduction the symptoms of various tick-borne diseases appear, but not once is there any information on either LD or LLD
Chapter 2.1 - the description of the handling of the sample is missing. The term according to the instructions for use, is too laconic.
Lines 159-161 - how were the samples stored?
Line 234 - what does repetaedly mean? How many times was the test repeated?
The text discusses the results, which are presented in the tables contained in the Supplementary file(s), which is very annoying. Since practically all the tables in the Supplementary file(s) are discussed in detail and are basically essential for understanding the results, this means that they should, or at least part of them, be in the main text. Certainly, table s6. The tables are incomparably more readable than the same information contained only in the text.
Reviewer 2 Report
Comments and Suggestions for Authors
This paper evaluated the TICKPLX multiplex ELISA at two laboratories (Australia and Finland) although the authors make some compelling arguments on the utility of this study, particularly in Australia where there is some controversy, the study design has some significant limitations. This should only be considered and presented as a preliminary/ proof of concept paper.
Most notable the samples size used for clinical validation and those shared between laboratories were small and lacked sufficient power to draw meaningful conclusions.
This was compounded by variation in sample preparation and dilution protocols that lack consistency. There was no statistical correction applied to account for this small sample size bias. This should be added or the sample size expanded. There was a lack of clinical outcome data that could help validate these test results, nor were other conditions considered that present in a similar fashion.
Only 5/28 markers were positive for the IgM antibodies, which actually suggests some non-specific reactivity. These should not be combined with the IgG results as a form of compensation. Afterall of the goal is to meaningfully evaluate TICKPLEX in Australia than appropriate sample size is needed that is not confounded by patient selection bias.
Overall the authors should provide more evidence on the clinical benefit and utility of these exams. As written, the authors do not provide justification that assays would improve patient outcomes or serve as cost effective alternatives. Given the potential for false positives a significant risk of overdiagnosis is possible and should be discussed. Additional details on how to interpret multi-pathogen results should be incorporated and would provide some clinical benefit which is lacking in this draft.
Overall the limitations should be expanded to explain some of the gaps on what was not expressed or can't be expressed because of these limitations. Alternatively this could be considered a pilot study with only preliminary data.
References should also be formatted consistently.
Reviewer 3 Report
Comments and Suggestions for Authors
Garg, K. at al.: Assessing interlaboratory performance and concordance of tick-borne disease testing using multiplex enzyme-linked immunosorbent assay V.1
This is a purely technical paper aimed at testing usability of a (Finish/European) commercial ELISA assay in a xeno-geographical (Australian) conditions. The study bases on a panel of 53 sera (only 20 of which from NS Wales) so dependability of the data is indeed limited (what the authors admit themselves). Nevertheless, the authors managed to extract a lot of valuable information of it with a high professionality. A bit strange circumstance - and a potential ethical issue - is an absence of any representant of the Australian staff - contributing to the sera collection as well as performing a part of the laboratory tests (p.3, l.120-2) - from the team of authors; nobody is mentioned personally in Acknowledgements, just "..the authors express gratitude to the Vector Borne Disease program at Royal North Shore Hospital for their assistance.." (p.14, l.599-600). That is an unfortunate omen as co-authorship of somebody detached would moderate understandable concerns about a bias when evaluation of the ELISA assay is (co)authored by shareholders of its manufacturing company... With this principal reservation, and conditionally on a necessary revision of the text, this manuscript is suitable for Pathogens.
This referee finds the article's title too general and little pertinent to what the authors, in fact, refer on. The keywords - such as in: 'Tickplex for testing Lyme-like cases in Australia' - should appear in the title...
The authors should check their tantamount wording - what may refresh a literary art piece, it may be counterproductive in scientific writing... Their, perhaps aesthetic, captivation with interchangeable terms manifests itself from the very start (Abstract, l.19-20: two equivalents, "TBD" and "TBI", alternate even within one sentence!) to the end of the transaction. Spontaneous jumping from one term to another of the same or almost the same meaning - e.g. a trifecta of synonyms defined on p.3.l.127: "TICKPLEX® PLUS (hereon referred to as TICKPLEX® or Index test or Index assay)..." - makes the text uneasy to understand to the reader. The authors should prune their palette of terms as much as possible.
P.1, l.28-31: an unfortune start of Introduction - note that (1) it is just and only the malaria quartette that can be regarded a "human vector-borne disease" in the full sense of the term - the other infections are rather zoonoses, and (2) the term "agent" commonly denotes a pathogen -> so something like this may be more correct: "ticks are the second most common blood-sucking parasites that transmit infectious agents to humans, the most widespread tick-borne disease (TBD) being LB..."
P.1. l.30-1: recalling a proposition given by EUCALB: be "Lyme disease" a term reserved for the American form of the disease (juvenile arthritis), then - to distinguish the European (Asian, etc.) disease's (multiaethiologic) manifestations - let us term them collectively "Lyme borreliosis" (or simply borreliosis). As a 'keeper of the EUCALB's heritage', this referee recommends to switch to the term LB in this context (and throughout the whole article)
P.1, l.32, and throughout the manuscript: Latin species names should be consistently written in italics
P.4, l.159: lower case in ¨It Is¨, pls.
P.4, l.178-9: "..Additionally, the R2 value should be greater than 0.75. An R-squared (R2), the coefficient of determination, was used.." note that usage of "R2" comes before its definition!
P.6, l.245-6: should read - ".. 0.81–1.00 indicates almost perfect to perfect agreement.."
Reviewer 4 Report
Comments and Suggestions for Authors
The manuscript titled “Assessing interlaboratory performance and concordance of tick-borne disease testing using multiplex enzyme-linked immunosorbent assay” aims to evaluate the ability of the multiplex TICKPLEX® assay to detect antibody responses to different aetiological agents of tick-borne diseases (TBDs), both in samples from patients with known infections in the Northern Hemisphere and in Australian patients with suspected TBIs/LLDs.
The TICKPLEX® test demonstrated high inter-laboratory reproducibility and potential diagnostic value for TBDs, including suspected Australian cases. In addition, the use of the test in Australia could help clarify the aetiological nature of LLD and improve the diagnosis of TBDs.
The study assesses the inter-laboratory concordance of the TICKPLEX® multiplex ELISA test on samples from patients with confirmed and suspected tick-borne diseases (including Australian cases of LLD). While the test shows good technical reproducibility, it presents significant critical issues such as low sample size: I) a low number of Australian samples (n=20), insufficient to draw generalisable conclusions on seroprevalence or aetiology of Lyme-like disease (LLD); II) australian samples were not uniformly characterized, some were already known to be seropositive for Borrelia, others were not; III) lack of a uniform reference standard limits the robustness of the comparative analysis; IV) the absence of a healthy Australian negative control group, which would have allowed the specificity of the test to be assessed in a non-endemic setting.
Furthermore, the lack of detailed clinical data (e.g. symptoms, duration, treatments received) makes it difficult to assess the clinical significance of observed antibody responses, and the exclusive use of serology in the absence of molecular data, such as PCR, limits the strength of conclusions. The evidence is insufficient to recommend publication in this Special Issue. Further validation on larger cohorts is needed to clarify the clinical relevance of the antibody responses observed. Further validation on larger, clinically and molecularly well-characterised cohorts is recommended before the study can be considered for publication.